# Gendered narratives and cultural shifts: A qualitative study on decadal changes in community alcohol consumption

Ming Gui Tan[1], Walton Wider[2], Nicholas Tze Ping Pang[3], Helen Benedict Lasimbang[3], Wendy Diana Shoesmith[3,4], Corine Rosapane M. Tangau[1], Leilei Jiang[5], Natchana Bhutasang[6]*

1 Department of Psychiatry and Psychological Health, Hospital Universiti Malaysia Sabah, Kota Kinabalu, Malaysia, 2 Faculty of Business and Communications, INTI International University, Nilai, Negeri Sembilan, Malaysia, 3 Faculty of Medicine and Health Sciences, Universiti Malaysia Sabah, Kota Kinabalu, Malaysia, 4 Derbyshire Healthcare NHS Foundation Trust, Trust Headquarters Kingsway Hospital, Derby, Derbyshire, England, 5 Faculty of Education and Liberal Arts, INTI International University, Nilai, Negeri Sembilan, Malaysia, 6 Faculty of Health Sciences, Shinawatra University, Pathum Thani, Thailand

* natchana.b@mru.ac.th

**Data Availability Statement:** https://doi.org/10.5281/zenodo.10813061.

## Abstract

Alcohol consumption has been a central practice in Sabah, Malaysia. However, this region has witnessed a nuanced shift in drinking habits over the last decade, raising concerns about the health, economic, and social implications of alcohol use within the community in Sabah. This study explores the impact of gender narratives and cultural transformations on alcohol consumption within Sabah over the last decade. The objectives of this research include 1) assessing the shifts in alcohol consumption patterns over the last 10 years, 2) understand the role of gender differences in shaping these patterns, and 3) identify the economic consequences resulting from the changes. Employing a qualitative approach, we conducted focus group interviews with members of the native community, each group consisting of up to 7 participants. Thematic analysis was used to identify key themes pertaining to gender roles, cultural practices, and socioeconomic influences of alcohol consumption. The data were then contextualized using Bronfenbrenner's social-ecological model and social role theory. The study reveals a significant shift in drinking habits. Historically, alcohol was mostly consumed by men, but now more women are drinking, changing long-standing gender roles related to alcohol use. Children often copy their parents' drinking behaviors, and with alcohol being more easily available, more people are drinking than before. Our findings also expose the complex consequences of alcohol use, which extend to health concerns, familial tension, and economic hardship. Despite the entrenched cultural status of alcohol, these negative outcomes are exacerbated by a lack of supportive healthcare services. In light of these insights, the study suggests the need for intervention plans that respect the cultural background of Sabah and account for gender dynamics while tackling the current issues of alcohol misuse. The research adds to the wider conversation about managing alcohol in different cultural settings and also recommended strategies based on the findings, such as cultural and gender sensitive community programs, youth centered

**Funding:** The author(s) received no specific funding for this work.

**Competing interests:** The authors have declared that no competing interests exist.

programs, community-based healthcare services, employment support and training and development of laws and policies.

## Introduction

Sabah, a state that exhibits notable cultural and economic differences from West Malaysia, is recognized as one of the leading regions in the country with a significantly elevated prevalence of alcohol consumption [1]. Reflecting a broader global trend, recent studies indicate a narrowing gender gap in alcohol use, with an increasing number of women engaging in alcohol consumption [2, 3]. This change is particularly relevant in Sabah, where its distinctive socio-cultural dynamics, shaped by the Kadazandusun community, play a significant role in alcohol consumption patterns [4]. Within this context, it is worth noting the significant prevalence of alcohol misuse, which is a pattern of alcohol consumption that leads to adverse health and social consequences [5, 6]. In the traditional celebrations of Sabah, alcohol consumption has been deeply ingrained throughout history [7]. This is especially true during the harvest festival known as the Kaamatan festival. This annual harvest ceremony holds great significance among the Kadazandusun community [8]. In this context, it is observed that the consumption of traditional home-brewed alcoholic beverages such as *tapai* and *montoku* holds significance, as they have been historically associated with various ceremonial occasions including the New Year, Christmas, birthdays, and routine social gatherings [9–11]. This paradoxical relationship between deep-rooted cultural practice and rising alcohol-related problems illustrates the lack of insight to the underlying driving forces.

However, while traditional and ceremonial practices celebrate alcohol as part of the cultural heritage, consumption patterns show a shift towards misuse. The consumption of alcohol is closely associated with cultural practices, particularly within indigenous communities [1]. For example, the Kadazandusun community is acknowledged for its distinctive patterns of alcohol consumption and ceremonies involving alcoholic beverages [12]. Despite the cultural emphasis on moderation, societal engagement, and spiritual obligations [4], the rising incidence of alcohol-related issues such as domestic violence, absenteeism, motor accidents, and physical confrontations remains concerning [6, 13]. Approximately 23.6% of alcohol consumption in Sabah can be classified as "risky," indicating the significant nature of this matter [14]. This paradox between the cultural practice and rising problems related to alcohol consumption highlights the insufficient understanding of sociocultural factors driving these changes.

Therefore, our study aims to unveil the complex relationship between tradition and alcohol-related issues, thereby laying down a strong foundation for future work to create culturally sensitive interventions. Firstly, we provide a literature review to give background on the role of alcohol in the history, and culture of Sabah and the theoretical framework employed throughout the study. We then detail our methods, focusing on how we used focus group discussions and thematic analysis to understand alcohol use. The findings section reveals the influence of gender and culture on how people in Sabah view and use alcohol. The objectives of this study include: 1) assessing the shifts in alcohol consumption patterns over the last 10 years, 2) understanding the role of gender differences in shaping these patterns, and 3) identifying the economic consequences resulting from these changes.

The importance of this study is twofold. On one hand, it offers insights to potential stakeholders to develop culturally appropriate programs in managing problems with alcohol misuse. On the other hand, this study contributes to the scientific community by offering a

broader and more comprehensive understanding of how cultural and gender influence alcohol consumption patterns.

## Literature review

Sociocultural, economic, and demographic factors are closely linked to the patterns of alcohol consumption in Sabah, Malaysia. In order to enhance our comprehension of these patterns, it is crucial to provide further details on the particular studies mentioned and critically evaluate their impacts on the field. Joseph et al. [10] highlighted the contrasting effects of alcohol by examining its influence on various aspects such as health, behavior, social dynamics, economy, and psychology. This study was utilized for its extensive analysis of the impacts of alcohol, which served as a basis for comprehending the complex aspects of alcohol consumption in Sabah. The authors demonstrated that moderate alcohol consumption can yield beneficial effects on both individuals and communities. However, while their research offers insights into the impact of alcohol consumption on the quality of life, the sample population may not be able to reflect the unique cultural landscape of Sabah. Furthermore, the study does not adequately consider the deep-seated cultural beliefs about alcohol that are prevalent in Sabah, which could significantly influence alcohol consumption patterns.

Alcohol plays a crucial role in the cultural and traditional practices of the indigenous communities in Sabah, serving as an essential component deeply woven into their social structure. In their study, Abd Rashid et al. [5] have identified a significant association between sex, religion, and obsessive-compulsive disorder with alcohol consumption in the background of increasing prevalence of alcohol use in several regions of Sabah. However, the study focuses narrowly on the role of religion without fully exploring the role of other cultural perspectives that may not be driven by religion, such as alcohol use in ceremonies and community norms about drinking.

Looking at other studies that address interventions in reducing alcohol-related problems, some studies like the one Shoesmith et al. [15] emphasized economic factors, such as taxation, as a primary approach to curb alcohol use. While this focus on regulatory measures through taxation may be helpful, it risks overlooking the cultural significance of alcohol in indigenous communities, which can backfire by fostering resistance among them. Moreover, Robert Lourdes et al. [1] and Mutalip et al. [14] included important demographic determinants such as gender, age, education, employment, and smoking habit in their analysis of alcohol consumption across Malaysia. Although they provide valuable insights into the general trends of alcohol consumption, the applicability of their findings to Sabah may be limited due to the diverse cultural perspectives on alcohol across different Malaysian states. Additionally, both groups of authors explore the cultural dynamics that influence alcohol consumption, which is a central element of our research. Regarding the general trend of alcohol consumption, Manickam et al. [16] and Mutalip et al. [17] have laid a good foundation by establishing the prevalence of alcohol use among young individuals. That being said, there remains a need to explore the factors motivating young people to consume alcohol, and whether these factors differ from those influencing adults.

As for the theoretical framework, this study employs Bronfenbrenner's social-ecological model alongside the social role theory. Bronfenbrenner's model provides a clear lens to view the pattern of alcohol use in Sabah across different levels–from the individual to broader societal impacts, including historical changes, which further facilitates the conceptualization of potential interventions [18]. Meanwhile, the social role theory helps to explore the narrowing gender gap in alcohol use as well as offering insights into the differing perceptions and behaviors between genders regarding alcohol consumption [19].

Collectively, these studies highlight a range of factors influencing alcohol consumption in Sabah but leave behind a common gap–the cultural and gender dynamics underlying the observations made by these studies are not thoroughly explored.

## Methodology

### Participants

The western coast is the focal point of the alcohol abuse epidemic due to cultural dynamics. However, due to the ongoing movement restriction in Malaysia amidst the COVID-19 pandemic, our target areas were confined to regions accessible from Kota Kinabalu. We engaged with village heads through networking to identify zones that agreed for interviews. Consequently, four zones, namely Kota Kinabalu (KK), Kudat, Penampang, and Kiulu, were selected as our target study areas, covering a vast geographical spread of 160km, thereby offering a diverse range of perspectives. The recruitment period started from 1 July 2020 to 31 August 2021.

Volunteer sampling was employed in recruiting participants, where anyone who was interested in discussing the alcohol consumption situation in the village and was at least 12 years of age was recruited. The age of 12 was used as a cutoff to strike a balance between the capacity to engage in extensive discussions and diversity of view. Participants less than 18 years old needed permission from their legal guardians to join the focus groups. They were also given reimbursement after completing the interview to incentivize participation.

One or more focus groups were formed in each target study area, each group consisting of 6 to 7 study participants. The youngest participant was 14 years old whereas the oldest was 81 years old. Our study involved 7 focus group discussions, comprising 6 to 7 members each.

### Procedure

Each focus group was supervised by a research associate accompanied by a facilitator. The facilitator helped to ensure clarity and cultural relevance using the local dialect on an as-needed basis. Additionally, the facilitator was responsible for recruiting participants and recording the sessions. At the start of each session, the objectives of the study and confidentiality protocols were clearly explained to participants, who then provided signed informed consent forms. For participants below 18 years of age, consent was obtained from their legal guardians. The group discussions took place in a community hall or similar common gathering area, ensuring privacy by not allowing unrelated personnel to be present.

Interviews were primarily conducted in Bahasa Malaysia, the nation's official language. However, the local dialect was occasionally employed to facilitate a deeper comprehension. Interviews typically lasted about 1.5 hours and were guided by questions developed through collaborative discussions with peers and stakeholders, which were:

1. How has alcohol consumption in your community evolved over the past decade?

2. How do gender and culture affect alcohol consumption in your community?

The facilitation process was designed to encourage open and honest dialogue, ensuring that all participants felt comfortable and valued by adopting a non-judgmental stance. Throughout the interview, efforts were made to encourage the expression of ideas and more extensive explanations. The facilitator also actively identified participants who may have been marginalized during the interview process and made attempts to ensure all participants were given an equal opportunity to voice their opinions.

Once the interviews concluded, participants were granted reimbursements in diverse forms such as food supplies, T-shirts, and face masks prepared in goodie bags. Participants who report alcohol use indicative of dependence or harmful patterns, or those exhibiting signs of potential mental health issues will be advised to seek evaluation at the nearest health clinics. Whereas for participants under 18 years old, their legal guardians will be notified and encouraged to facilitate an assessment.

## Reflexivity

Several characteristics of the main researcher may color his view on this topic. The main researcher, a healthcare practitioner specializing in psychiatry with an urban background, and having influenced by Buddhist philosophy, initially approached the study of alcohol consumption primarily as a coping mechanism. This perspective was challenged during the research, especially when encountering the perceived paradox of the intertwining "good" and "bad" aspects of alcohol, such as its recreational use (bad) as part of family bonding (good).

Throughout the coding process, the main researcher noticed an initial tendency to focus extensively on the negative impacts of alcohol. Reflecting on this bias, deliberate efforts were made to reconsider how the role of alcohol was conceptualized. This involved iterative revisions of the coding framework to ensure a more balanced representation of the impact of alcohol use. Regular discussions and debates with peers during the coding and theme formation stages also helped to enhance the neutrality of the conceptualization.

Interpersonally, the dynamics within the interview settings also required careful navigation. Firstly, the researcher acknowledged the potential power imbalance between the researchers and participants, which might lead them to provide responses they thought were expected. Furthermore, variations in religious beliefs, age, and gender among participants may have influenced the power dynamics within the group, especially during discussions on controversial topics such as underage drinking. It was challenging to balance between having a diverse group which could introduce complex power dynamics that might hinder open discussions, and having a homogenous group (such as same gender or similar age range) which risked creating an echo chamber. Some of the strategies used to address these challenges include maintaining reflective diaries to document and manage potential biases, particularly during the coding phase. Regular discussions with supervisors provided external perspectives, ensuring a broader and neutral interpretation of the data.

## Ethical considerations

During our qualitative research, we recognized that the close-knit nature of the village communities may lead to participants withholding information due to the presence of other community members, affecting the depth of their responses. To address these concerns and ensure the accuracy of our data, we implemented strict measures to protect participants' identities and the confidentiality of the discussions. Data anonymization procedures included the use of pseudonyms for participants and the removal of identifiable information from transcripts. We used alphabetical codes (A to G) and group numbers (1 to 7). Specific details about locations or other identifying information were intentionally left out. By taking these measures, we aimed to build trust and ensure participants felt comfortable sharing their genuine experiences. All data were stored securely on password-protected devices to ensure confidentiality. After transcribing and verifying the accuracies of the audio recordings, they were destroyed from all devices.

Ethical approval was obtained from the ethics committee of the Universiti Malaysia Sabah.

## Data analysis

Following the six-phase process proposed by Braun and Clarke [20], our thematic analysis began with the transcription of audio recordings from participant interviews. A third-party assistant with specialized training handled this transcription, aiming for both precision and an impartial reflection of participants' statements to minimize potential biases at the outset. Upon acquiring the transcribed data, our analysis proceeded through manual thematic analysis, supported by ChatGPT for efficient data organization.

The researcher began by listening to the audio recordings and repeatedly reading through the transcriptions to become deeply familiar with the content. This phase involved noting inflections, breaks, and other nuances to fully understand the data.

During the code development phase, a hybrid model of coding (inductive and deductive) was employed. Initially, some codes such as "pandemic" and "tradition" were predefined. As the analysis progressed, additional codes like "selection" emerged from the data. ChatGPT was utilized to identify and tag contextually relevant keywords corresponding to these codes, facilitating the retrieval process and allowing the researcher to efficiently highlight relevant quotations within the transcripts.

Once coding was completed, the focus shifted to developing themes by interpreting and organizing codes into meaningful groups. For instance, the codes "motivating," initially defined as "factors or reasons that encourage the initiation or continuation of drinking," and "embrace," defined as "views that acknowledge both positive and negative aspects of alcohol," were merged into a single code-named "motivating factors." This new code encompassed factors, reasons, and views that support the initiation or continuation of drinking.

Through iterative discussions within the research team, patterns were identified, and candidate themes were actively constructed to address the research questions. These themes were refined and finalized after a thorough review process. An expert review process was integral to evaluating the themes, involving feedback from seasoned researchers and representatives of the studied community.

In the conceptualization phase, themes were examined through the lens of Bronfenbrenner's social-ecological model and social role theory. This approach explored the immediate social contexts of alcohol use (microsystem) and extended implications such as community norms (exosystem). The social role theory provided insight into how participants' views and behaviors are shaped by their social roles, with a focus on the relationship between gender roles and drinking behaviors.

Finally, all themes were consolidated and examined through the theoretical lens, and a report was put together to clearly describe our findings. The tagging and retrieval capabilities of ChatGPT ensured that the analysis was efficient and thorough, allowing for detailed quotations to be identified and discussed to maintain objectivity and neutrality. ChatGPT provides leverage in managing large volume of data quickly. However, we recognise its limitations such as the potential for misinterpretation of certain context, figure of speech and nuances in human language. Therefore, peer debriefing sessions and peer discussions provided an additional layer of scrutiny, reinforcing the reliability and credibility of our findings.

## Results

Based on all the responses gathered, four themes emerged.

### Tradition and the shifting role of alcohol

The consumption of alcohol in our community is deeply entangled with traditions and changing perceptions. Throughout history, alcohol has played a pivotal role in celebrations and rites

of passage, a sentiment echoed by a respondent who remarked on the inescapability of tradition, ". . .cannot escape tradition. . ." (B6). However, contemporary narratives suggest a shift from these traditional moorings. One respondent captured this evolution, stating, "It feels like the traditional meaning has been lost now. People used to drink to celebrate, but now it's more to run away from problems." (F1). This indicates a profound change in the role and significance of alcohol within the community—from a symbol of joy and communal bonding to a means of escape from personal and societal issues. The entrenched status of alcohol within their culture is further demonstrated by the prevailing sense of disappointment over the waning tradition of alcohol brewing. For example, the Kaamatan festival that is traditionally associated with religious celebration has seen shifts from traditional *tapai* consumption to a more commercialized and less ritualistic drinking pattern.

The COVID-19 pandemic has further complicated the prevalence of alcohol use, exerting dual effects influenced by various factors. While movement restrictions imposed by the local government might have limited access to alcohol, thereby reducing consumption for some, the accompanying economic decline and increased social isolation have conversely led to an increase in alcohol consumption for others. A participant explained, "The use of alcohol has changed from the past because there was no COVID-19 pandemic before; people spent time going to work to earn a living. Now, many are out of work, many don't know what to do, and they drink alcohol to kill time" (F1). This reflection not only demonstrates the impact of the pandemic on daily life and employment but also highlights the transition of alcohol use towards the role of a coping mechanism.

Moreover, the analysis revealed a concerning trend of underage drinking. Although participants under 18 denied consuming alcohol, adults recalled their first experiences with alcohol at ages as young as 8 or 9. Traditionally, alcohol consumption in the community served ritualistic purposes. However, recent trends indicate a shift toward non-ceremonial drinking at increasingly younger ages. A significant factor contributing to early exposure to alcohol is parental influence. Although typically, parents act as deterrents to early alcohol use, from our observation, we notice that parents could also introduce their children to alcohol. For instance, one participant recalled, "I started drinking as young as 10 when I followed my parents to neighbours' houses" (D4). Additionally, there appears to be a relaxation in the traditional supervisory roles of parents, reflected by a participant narrating how he used to drink behind his parents' backs. This phenomenon reflects not only a change in family dynamics and supervision, but also a shift of the role of alcohol from regulated use in ceremonies to a more casual use in other common situations.

## The double-edged sword of visibility and accessibility

The alcohol availability in our community has experienced substantial changes, as highlighted by a respondent's observation: "In this place, there are no obstacles or prohibitions to buying alcohol" (E2), illustrating the ease with which alcohol can now be purchased. This ease of access extends beyond commercially sold alcohol to include local homebrewing. As one participant described, "Here, we used to make our alcohol using rice and medicine, and it had to be made for a week before it could be consumed" (A3). This traditional homebrewing process, while requiring specific ingredients and significant time, contrasts with the readily available commercial alcohol, which offers consistent year-round availability. Through this deeply rooted cultural tradition, the mood of pride and self-reliance could be observed. Nevertheless, in some regions, traditional brews remain the preferred choice due to the ease of storing ingredients and alcohol. This preference is especially noticed in rural areas where commercial alcohol is less accessible. A participant reported, ". . .every house must have a stock of alcohol,

especially in village areas, because this alcohol can last for several months. . .” (A1). Further-more, “. . .*tapai* is easily available because it is made from ingredients like cassava or rice. Mod-ern canned drinks are hard to find now unless you go to the city. . .” (A1). These observations reveal a seemingly contradictory yet fundamentally similar theme–accessibility, where each phenomenon ultimately caters to the need for accessible alcohol.

The transition from scarcity to abundance is further reflected by another respondent, who noted, "It used to be hard to find stores that sold alcohol. Now they are almost on every corner" (A2). This change points to a dramatic increase in the physical visibility of alcohol, making it a commonplace feature in the Sabahan community. Echoing this sentiment, another respondent adds, "Alcohol is cheaper and easier to find compared to ten years ago" (F7), indicating not only increased availability but also affordability over time. The ubiquity of alcohol is captured in the routine experiences of community members, as one person remarked, ". . .we often go into Speedmart 99. When we go in, we see it there." (E6). These statements reflect how alcohol has seamlessly integrated into daily life and consumer habits.

Collectively, the increasing accessibility and affordability of alcohol may lead to higher rates of misuse and dependence within the community. Furthermore, the ease of obtaining alcohol, combined with existing behavioral reinforcements such as peer influence and stress relief, raises concerns about promoting alcohol consumption among younger individuals below 18 years old. This trend is particularly alarming as underage drinking is not only prevalent but often occurs in contexts where alcohol is seen as an acceptable part of social interactions. For example, young people may start drinking under the guise of cultural traditions, which can obscure the severity of early alcohol exposure.

## Social influences and evolving gender norms in alcohol consumption

The patterns of alcohol consumption in our community are greatly influenced by peer pres-sure and changing gender norms, which demonstrate the interaction of social constructs. One respondent encapsulated the peer pressure phenomenon among the youth, stating, "Some-times, when consumed among teenagers, it is referred to as ’psycho’, wanting to show oneself as someone strong" (G1). This desire to show their bravado illustrates how young individuals use alcohol to conform to peer expectations. The pressure to conform extends to adults as well, another participant added, ". . .if we don’t drink, they will ignore us" (A2), it is obvious that there can be social consequences for abstaining. Beyond peer influence, there is also a strong cultural pressure stemming from longstanding traditions to partake in alcohol consumption. This cultural expectation is so ingrained that refusing to drink can sometimes be seen as going against communal norms.

Family influence also plays a pivotal role in shaping attitudes and behaviors toward alcohol. The narrative, "If parents drink, it’s highly likely their children will also drink" (A2), points to the generational transmission of drinking habits, underscoring the impact of parental behavior on children. This notion of familial legacy in alcohol consumption is further supported by another participant, who observed, ". . .it’s been from long ago, from parents, grandparents, they indeed drank. So, our generation now, follow in their footsteps" (E3). Such observations highlight how family traditions contribute to the continuity of alcohol consumption across generation, which not only sustains traditional practices, but also perpetuates potentially harmful behaviors associated with alcohol use.

A significant shift in gender dynamics shows increasing active participation from women in alcohol consumption, traditionally dominated by men. "I remember before only men drank. Now, I see many women also drinking" (A4), one respondent remarked, as well as "Before, I thought only men could make alcohol at home. Now, I see many women also trying

to make it themselves" (C4). Despite these advancements, societal judgments persist, with a discernible bias in how male and female drinkers are perceived, "There's a difference in our society. If a man drinks, people say he's strong. But if a woman drinks, they look down on her" (B4). This dichotomy suggests that while progress has been made, deep-seated cultural norms still influence the acceptance of female alcohol consumption. Furthermore, according to one of the participants, ". . .because when her husband drinks, the woman automatically takes care of him, especially if he is around 40 or 50 years old. . ." (B5), suggests that the traditional supportive role of women had contributed to male dominance in alcohol consumption.

## The multifaceted impact of alcohol consumption

The consumption of alcohol has wide-ranging repercussions, ranging from health, relationships, work, to financial stability. Often beginning with experimentation in youth, the journey into long-term alcohol use can bring about negative implications. One participant reflected, "I used to drink to forget my problems. But now I realize it only adds to the problems" (D2), indicating a recognition of the detrimental effects of alcohol use later in life and the ineffectiveness of alcohol consumption as a coping mechanism.

On a personal level, health deterioration due to the direct and indirect effects of longstanding alcohol use is a significant concern elicited from the group discussions. As stated by a respondent, "I try to stop drinking because I feel it affects my health" (D2), this demonstrates a conscious decision to reduce health risks by decreasing alcohol intake. Imbalanced food intake during intoxication and an increased speed of aging are some health concerns reflected by participants. Addiction, ranging from dependence to severe misuse, is another crucial topic throughout all target zones. On a lesser level, some individuals were willing to spend more on obtaining alcohol than on any other necessities in life. However, in extreme cases, people have brought physical harm to themselves while intoxicated, including ingesting poison due to exacerbated despair and involvement in road traffic accidents.

The negative effects of alcohol extend beyond personal health to interpersonal relationships. Alcohol consumption has strained family bonds, as encapsulated by one participant, "I feel drinking alcohol can damage health and family relationships" (D7). Marital conflicts often arise with the use of alcohol, either through exacerbating pre-existing tensions or creating new conflicts: ". . .they would bring up old matters and quarrel about that when they are drunk. . ." (A2). These marital conflicts are not limited to the couples but also affect their children, leading to emotional trauma and physical displacement from home until the dispute is resolved. Additionally, interracial tensions can be heightened through alcohol-mediated disinhibition, contributing to physical fights.

The workplace is also affected by alcohol use, evidenced by accounts of job losses attributable to drinking, ". . .lost their job because they got drunk at work. . ." (F1). This phenomenon, in the context of the COVID-19 pandemic facilitating alcohol use, could have a double negative effect on occupational settings. Furthermore, financial strain is another concern related to alcohol use. The shift from spending on alcohol to prioritizing family finances is evident in the reflection of one participant: "I used to spend a lot of money on alcohol. Now I try to save that money for my family" (D7). This shift in perspective demonstrates a growing awareness of the financial toll of excessive alcohol consumption.

While these personal experiences highlight the negative consequences of alcohol on health, relationships, employment, and finances, it is essential to explore the broader societal factors contributing to these impacts. Determinants such as socioeconomic disparities, access to healthcare, cultural norms, alcohol marketing practices, regulatory policies, and healthcare interventions play a significant role in influencing consumption patterns and outcomes.

## Discussion

Previous studies mentioned in the literature review section have succinctly demonstrated the prevalence of alcohol consumption and patterns of consumption, examined how religion and culture affect alcohol use, and explored how economic factors like taxation can influence alcohol consumption. While the findings of this study align with existing literature on the significant role of cultural traditions and social pressures in shaping alcohol consumption patterns, they shed light on the complex interplay of underpinning sociocultural and gender-related dynamics.

Historically, alcohol has been deeply ingrained in the cultural fabric of this community, playing a central role in ceremonies and celebrations, akin to many indigenous societies globally [21]. Traditional alcoholic beverages like *tapai*, *lihing*, *tumpung*, and bahar have symbolized communal unity and ancestral customs, though their significance has evolved over time, with alcohol now serving as a coping mechanism for some individuals amidst life's challenges [22, 23]. This transition from traditional to more problematic uses of alcohol challenges the notion that cultural traditions solely act as a protective factor against alcohol misuse [5].

The effects of global events such as the COVID-19 pandemic and the subsequent Movement Control Order (MCO) on consumption patterns were seemingly contradictory. While urban areas experienced a reduction in alcohol consumption due to restrictions on mass gatherings, rural regions witnessed a continuation of traditional drinking habits, indicating a geographical shift in consumption patterns [24]. These changes reflect broader global trends where societal stressors drive individuals towards substance use [24, 25]. This relates to our findings, which show how accessibility to urban areas and availability of alcohol production mediate the impact of the pandemic on consumption patterns. Our study supports the findings by Shoesmith et al. [15] that economic limitations are closely associated with alcohol use. Furthermore, the availability and accessibility of alcohol highlighted in our research coincides with the findings of Robert Lourdes et al. [1].

The discussion also highlights the influence of age, urbanization, and beverage preferences on alcohol consumption. Older generations in rural areas tend to favor traditional homemade beverages, whereas younger demographics in urban settings lean towards commercially available alcoholic drinks [26]. Youth involvement in alcohol consumption is concerning, with peer influence and societal pressures driving premature and potentially hazardous drinking habits [27]. Family dynamics and community regulations also shape alcohol-related behaviors, with parental attitudes towards alcohol influencing offspring in complex ways [28]. For instance, the intergenerational transmission of drinking habits through cultural ceremonies highlights the significance of alcohol in maintaining social cohesion. Although unique local traditions, such as the brewing and consumption of *tapai* and *lihing*, continue to influence drinking patterns, they coexist with the convenience and affordability of commercially available alcohol, reflecting a blend of old and new approaches to maintaining the cultural heritage. Additionally, the socioeconomic disparities brought about by urbanization, such as financial hardship and employment instability, play a significant role in shaping alcohol consumption as a coping mechanism. These findings echo the findings of the research by Joseph [29] that highlighted the role of family influence in the continuation of drinking habits.

Furthermore, the evolving gender dynamics in alcohol consumption reveal a transition towards increased active participation of women, reflecting broader shifts in gender roles [5]. This shift can be attributed to several driving factors, including increased socioeconomic and educational opportunities for women, empowering women to depart from their passive roles in a community. Despite these changes, we observed the lingering presence of gender biases and societal judgments that continue to challenge the transitioning role of women. This

uncovers the ongoing tension between traditional norms and contemporary practices. These findings illustrate the complex gender dynamics in alcohol consumption and emphasize the need for targeted interventions to promote gender equality and cultural acceptance.

The detrimental effects of alcohol on health, interpersonal relationships, and risky behaviors shows the need for more comprehensive interventions, including educational counselling services on moderate alcohol use [30, 31]. Despite the inadequacy of health services related to alcohol, initiatives driven by religious affiliations have dominated legislative discussions, indicating a gap in addressing core issues [31]. Our results further reinforce the findings of previous studies by demonstrating the combined impact of traditional and modern alcohol consumption practices on health issues. [1, 14] Overall, the discussion emphasizes the multifaceted nature of alcohol consumption within the Sabah community and the need for tailored interventions to address its cultural, social, and health-related dimensions.

In terms of theoretical framework, Bronfenbrenner's social-ecological model sheds light on how different social contexts, from individual experiences to broader societal influences, influence alcohol consumption patterns. It allows us to view the layers of influence that shape drinking habits, namely the microsystem, mesosystem, exosystem, macrosystem and chronosystem. The microsystem denotes the immediate environment of the individual, including family and peer influence. Our study highlights the effects of parental drinking on younger individuals, and also pointed out the role of peer pressure in early exposure to alcohol use. The interactions between family, peer groups and the community forms the mesosystem that changes an individual's approach to alcohol use, such as community activities that reinforce drinking habits at home. As for the exosystem, the accessibility of alcohol, especially in rural areas where homebrewing is common, as well as local policies on alcohol sales play a crucial role in maintaining drinking habits. Simultaneously, economic hardships exacerbated by the pandemic also led to a heterogenous pattern of alcohol use to individuals from different communities. Cultural traditions that are deeply ingrained in the Sabahan community make up the macrosystem, which are also ever evolving and in the downstream altering social acceptance and patterns of alcohol use. The macrosystem also consists of evolving gender dynamics that reflect broader changes, with increased active participation of women in alcohol use, challenging traditional gender roles. Lastly, the chronosystem, representing changes over time, is reflected through the shifts driven by the COVID-19 pandemic and generational shifts.

Besides the social-ecological model, the study is also viewed through the lens of social role theory to explain how changing gender roles in the community affect both drinking patterns and perceptions towards alcohol use. In our study, we observe a significant shift in the traditional roles of women. Although driving force behind this shift was not explored, one hypothesis is that there is increased socioeconomic and educational opportunities for women, allowing them to step away from their passive roles within the community. Nevertheless, despite these advancements, societal judgments persist, in particularly how the drinking behaviors of men and women are interpreted within the community. This ongoing tension between traditional societal norms and contemporary changes illustrates the complex gender dynamics in alcohol consumption. By using social role theory, we are able to elucidate how evolving gender roles reshape alcohol consumption patterns and perceptions within the Sabahan community.

## Limitations and recommendations for future research

The research presented primarily focuses on the western coast of the Sabah community, which might not capture the entirety of practices and perspectives prevalent in other regions within Sabah. This geographical focus presents a limitation in understanding the full depth and breadth of alcohol consumption patterns within the broader Sabah community. Moreover, the

data captured in the study is up to the present time, therefore, given the fluid nature of societal perspectives and practices, the findings might not encapsulate potential future shifts.

Another significant consideration is the background of the researcher, both as a healthcare staff member and in terms of religious beliefs. These may have introduced a potential bias in the way participants responded, as they might have provided answers they deemed 'acceptable' or 'expected' within a healthcare context. This could particularly influence responses related to alcohol consumption patterns and their consequences.

Additionally, while the study emphasizes the historical and cultural importance of alcohol to the community, there might be limitations in the depth of exploration concerning specific rites, rituals, and traditions surrounding its use. Addressing the sensitive aspects of the research, especially around gender dynamics might also have been influenced by reporting biases. Furthermore, the exploration of the impacts of alcohol was not comprehensive, potentially limiting the depth of understanding regarding the full scope of alcohol-related consequences in the community.

For a more comprehensive understanding of the relationship of the Sabah community with alcohol, it would be beneficial to expand the geographical scope of future studies. This would ensure a richer understanding by capturing diverse views within Sabah. A longitudinal approach could provide invaluable insights into the evolving patterns of alcohol consumption and its broader implications.

Given the potential bias introduced by the researcher's background, future studies should involve a diverse team of researchers and implement more rigorous peer discussions to ensure a neutral data collection process. This approach could alleviate participants' concerns and encourage more candid responses.

A dedicated focus on the specific cultural and ceremonial uses of alcohol might offer a deeper dive into its traditional significance. Collaborating across disciplines with anthropologists, sociologists, and health professionals can paint a more holistic picture.

Lastly, the alarming trends among the youth that have been highlighted in the findings suggest a need for more in-depth studies targeting this demographic.

## Conclusion

The narratives from the Sabah community reveal a complex interplay in the dynamics of alcohol consumption. While the cultural significance of alcohol consumption remains undeniable, seismic shifts are observed in the relationship of consumption with perceptions, evolving gender roles, and global events. As these influences intersect with local traditions, the emerging challenges are complex and require approaches that respect deep-rooted traditions, account for gender dynamics, and address contemporary issues.

Based on our findings in the study, we recommend a multifaceted approach to address the issues surrounding alcohol consumption in the Sabah community, which includes:

### Cultural and gender sensitive community programs

These campaigns should be tailored to the cultural context of Sabah and leverage on local languages and cultural narratives to tackle alcohol misuse in a more relatable manner. Stories and testimonies of the harm of alcohol misuse and success of handling alcohol-related issues could be incorporated into these programs on top of didactic teachings to personalize the message. We also recommend promoting awareness about the negative implications of alcohol consumption through health programs, focusing on moderation instead of total abstinence from alcohol. These programs could be developed in collaboration with local non-governmental organizations, utilizing locals to approach their communities and thereby reducing resistance

from villagers. Additionally, addressing the gender dynamics uncovered in this study calls for more gender-sensitive interventions, focusing on both the empowerment and protection of women against societal judgments.

### Youth centered programs

Establish and reinforce pre-existing programs in schools and community centers aimed at preventing underage drinking. Strategies like engaging parents and caregivers in workshops to educate them on the impact of adult drinking behaviors on their children should also be considered. We also recommend engaging with youth in a wider variety of community-based programs to potentially replace alcohol misuse as a coping mechanism and social cohesion amongst peers.

### Community-based healthcare services

As for the role of healthcare services, interventions for alcohol misuse should be prioritized, particularly in rural areas. These interventions should focus on managing alcohol dependence, addressing alcohol-related mental and physical health issues, and providing family-centered programs aimed at breaking the cycle of intergenerational transmission of drinking habits.

### Employment support and training

Supported employment has been an essential component in the treatment of people with mental illness, there is an existing guideline on implementation supported employment programs in Malaysia. In managing individuals with alcohol misuse, job training and economic support programs may be employed to address the financial hardships that drive their drinking patterns, which could potentially reduce the reliance on alcohol as a coping mechanism.

### Development of laws and policies

Regulations on the sale of alcohol should be reviewed and strengthened, particularly in rural areas where homebrewing is prevalent. The existing laws regarding the sale of alcohol to minors should also see increased enforcement to reduce accessibility and availability of alcohol to minors. At the policy development level, it is important to implement stricter regulations targeting high-risk groups, such as youth, while adopting a more relaxed approach for adults, aiming to reduce the prevalence of underage drinking while recognizing the autonomy of adults to drink in moderation.

While our study has explored the complexities of alcohol consumption patterns and socio-cultural factors, there remains a need for future research to delve into the specific patterns identified. Future studies can build on our findings and contribute to a more comprehensive understanding of alcohol consumption patterns, such as through longitudinal studies to track the shifts we have observed or by investigating the impact of gender dynamics, including domestic violence. These efforts will enhance both the effectiveness of interventions and the well-being of the Sabahan community.

## Author Contributions

**Conceptualization:** Helen Benedict Lasimbang, Wendy Diana Shoesmith.

**Formal analysis:** Ming Gui Tan, Walton Wider, Wendy Diana Shoesmith.

**Investigation:** Ming Gui Tan, Wendy Diana Shoesmith, Corine Rosapane M. Tangau.

**Methodology:** Nicholas Tze Ping Pang, Wendy Diana Shoesmith.

**Project administration:** Helen Benedict Lasimbang.

**Software:** Wendy Diana Shoesmith.

**Supervision:** Walton Wider, Helen Benedict Lasimbang, Wendy Diana Shoesmith.

**Writing – original draft:** Ming Gui Tan, Walton Wider.

**Writing – review & editing:** Walton Wider, Nicholas Tze Ping Pang, Helen Benedict Lasimbang, Wendy Diana Shoesmith, Leilei Jiang, Natchana Bhutasang.

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
