## [Decision Letter · Decision Letter 0]

22 Feb 2024

PONE-D-23-38278Gendered narratives and cultural shifts: a qualitative study on decadal changes in community alcohol consumptionPLOS ONE

Dear Dr. Tan,

Thank you for submitting your manuscript to PLOS ONE. After careful consideration, we feel that it has merit but does not fully meet PLOS ONE’s publication criteria as it currently stands. Therefore, we invite you to submit a revised version of the manuscript that addresses the points raised during the review process. The manuscript has been assessed by 2 reviewers and their comments are available below. The reviewers have raised some major concerns. They feel the manuscript would benefit from greater structure and detail in the abstract, stronger literature review and study rationale and elaboration of methodology and data analysis and discussion. Could you please carefully revise the manuscript to address all comments raised?

We look forward to receiving your revised manuscript.

Kind regards,

Annesha Sil, PhD

Associate Editor, PLOS ONE

Reviewers' comments:

Reviewer's Responses to Questions

**Comments to the Author**

1. Is the manuscript technically sound, and do the data support the conclusions?

Reviewer #1: Yes

Reviewer #2: Partly

2. Has the statistical analysis been performed appropriately and rigorously? 

Reviewer #1: No

Reviewer #2: N/A

3. Have the authors made all data underlying the findings in their manuscript fully available?

Reviewer #1: Yes

Reviewer #2: No

4. Is the manuscript presented in an intelligible fashion and written in standard English?

Reviewer #1: Yes

Reviewer #2: Yes

5. Review Comments to the Author

Reviewer #1: My only comment is that you review the form of citation of the document, since they must be surnames and in some citations they have names, just verify according to the editorial guidelines. It is a good article and of great importance for theoretical bases of this problem.

Reviewer #2: This manuscript on gendered narratives and cultural shifts in alcohol consumption in the Sabah community has some potential. However, I found it wasn’t guided by a clear research question and was, at times, a little hard to follow. I’m also not sure this is the most appropriate journal for it (perhaps a journal more oriented sociological/cultural studies and substance use). Below I’ve listed some of the more substantial issues I found with the paper:

• I found the abstract difficult to follow – it jumped around without contextualising Sabah or providing a clear research question. More detail is needed here, and I would suggest a more structured abstract based off the introduction, methods, results and conclusion might be helpful.

• The structure of the paper needs to be outlined much earlier in the introduction, which I think will help. While the information is there, the current structure doesn’t always flow very coherently between topics and ideas

• You state the “study seeks to examine the trajectory of alcohol consumption within the Sabah community over the years” but I’m not sure you can do this with a study of only one timepoint. Perhaps it might be better to phrase this as reflections of cultural and gender shifts in alcohol consumption in Sabah

• The studies raised in the literature need elaboration – what did the studies do and why did you draw on them? Some of the studies seem to be focused on Sabah but others are drawn from Malaysia more broadly. What can/can’t these studies tell us? Where is the research gap you are filling?

• The methods are generally ok, but going through the COREQ checklist may help clarify some things. Importantly, the “Data Analysis” section is very underdone. How was the data “processed manually”? Was thematic analysis used? If so, how was it used? I’m also a little unsure about the use of ChatGPT as a tool to assist with analysis given the potential for bias/inaccuracies

• I found many of the themes too brief to provide substantial analysis and interpretation. Lots of words were used up writing out both translations of quotes (authors should decide if this is worth the wordcount used) and I didn’t feel like the themes told enough of a coherent story. I would consider collapsing themes if/where possible, but also using a reformulating the themes so they don’t read like lists of quotes from participants

• I felt the discussion spent a lot of time reviewing the literature, but didn’t incorporate the findings to draw out comparisons/differences from previous research. A stronger discussion that draws on both relevant literature alongside your findings would provide clearer implications/conclusions for the paper.

6. PLOS authors have the option to publish the peer review history of their article (what does this mean?). If published, this will include your full peer review and any attached files.

Reviewer #1: No

Reviewer #2: No

---

## [Author Response · Author response to Decision Letter 0]

11 Apr 2024

Reviewer 1:

My only comment is that you review the form of citation of the document, since they must be surnames and in some citations they have names, just verify according to the editorial guidelines. It is a good article and of great importance for theoretical bases of this problem. 

Response: Thank you for the comments. We have revised accordingly.

Reviewer 2:

This manuscript on gendered narratives and cultural shifts in alcohol consumption in the Sabah community has some potential. However, I found it wasn’t guided by a clear research question

Response: We have made amendments and came up with a stronger research question.

I’m also not sure this is the most appropriate journal for it (perhaps a journal more oriented sociological/cultural studies and substance use)

Response: We feel that PLOS One is a suitable journal, as alcohol use is a very interdimensional and intertextual disorder that cuts across medicine, psychology, sociology and gender, hence we do not want to limit the scope by sending it to a very specific journal, but rather to a broad based journal.

I found the abstract difficult to follow – it jumped around without contextualising Sabah or providing a clear research question. More detail is needed here, and I would suggest a more structured abstract based off the introduction, methods, results and conclusion might be helpful.

Response: We have made necessary amendments according to this comment.

The structure of the paper needs to be outlined much earlier in the introduction, which I think will help. While the information is there, the current structure doesn’t always flow very coherently between topics and ideas

Response: We have restructured the introduction to present the outline clearly.

You state the “study seeks to examine the trajectory of alcohol consumption within the Sabah community over the years” but I’m not sure you can do this with a study of only one timepoint. Perhaps it might be better to phrase this as reflections of cultural and gender shifts in alcohol consumption in Sabah

Response: This has been amended.

The studies raised in the literature need elaboration – what did the studies do and why did you draw on them? Some of the studies seem to be focused on Sabah but others are drawn from Malaysia more broadly. What can/can’t these studies tell us? Where is the research gap you are filling?

Response: We've revised the paper to provide more details on cited studies, clarified their distinctions, and emphasized our contribution to filling the research gap

The methods are generally ok, but going through the COREQ checklist may help clarify some things. Importantly, the “Data Analysis” section is very underdone. How was the data “processed manually”? Was thematic analysis used? If so, how was it used? I’m also a little unsure about the use of ChatGPT as a tool to assist with analysis given the potential for bias/inaccuracies

Response: We reviewed the COREQ checklist to ensure clarity in our methods. We acknowledged the need for further detail in the "Data Analysis" section and elaborated on how the data was processed manually through thematic analysis. Regarding the use of ChatGPT, we understood your concerns about bias and inaccuracies and provided additional justification for its inclusion or considered alternative methods.

I found many of the themes too brief to provide substantial analysis and interpretation. Lots of words were used up writing out both translations of quotes (authors should decide if this is worth the wordcount used) and I didn’t feel like the themes told enough of a coherent story. I would consider collapsing themes if/where possible, but also using a reformulating the themes so they don’t read like lists of quotes from participants

Response: Thank you for your feedback. We have reviewed the themes and acknowledged the brevity you mentioned. We reconsidered the use of wordcount on translations of quotes and made adjustments. We collapsed themes where possible and reformulated them to ensure they provide a more coherent narrative, rather than reading like lists of participant quotes.

I felt the discussion spent a lot of time reviewing the literature, but didn’t incorporate the findings to draw out comparisons/differences from previous research. A stronger discussion that draws on both relevant literature alongside your findings would provide clearer implications/conclusions for the paper.

Response: We have revised the discussion so that it is now more concise now and draws out comparisons/differences from previous research.

---

## [Decision Letter · Decision Letter 1]

1 May 2024

PONE-D-23-38278R1Gendered narratives and cultural shifts: a qualitative study on decadal changes in community alcohol consumptionPLOS ONE

Dear Dr. Bhutasang,

Thank you for submitting your manuscript to PLOS ONE. After careful consideration, we feel that it has merit but does not fully meet PLOS ONE’s publication criteria as it currently stands. Therefore, we invite you to submit a revised version of the manuscript that addresses the points raised during the review process.

We look forward to receiving your revised manuscript.

Kind regards,

Gabriel Caluzzi

Guest Editor

PLOS ONE

Additional Editor Comments:

The reviewers (who I note are new to the manuscript and reviewing it for the first time) are largely supportive, and have suggested a number of specific and general areas for revision. This includes suggestions for synthesis of the literature are some more detail in the analysis, discussion, and methodology sections.

Reviewers' comments:

Reviewer's Responses to Questions

**Comments to the Author**

1. If the authors have adequately addressed your comments raised in a previous round of review and you feel that this manuscript is now acceptable for publication, you may indicate that here to bypass the “Comments to the Author” section, enter your conflict of interest statement in the “Confidential to Editor” section, and submit your "Accept" recommendation.

Reviewer #3: (No Response)

Reviewer #4: All comments have been addressed

2. Is the manuscript technically sound, and do the data support the conclusions?

Reviewer #3: Partly

Reviewer #4: Partly

3. Has the statistical analysis been performed appropriately and rigorously? 

Reviewer #3: I Don't Know

Reviewer #4: N/A

4. Have the authors made all data underlying the findings in their manuscript fully available?

Reviewer #3: No

Reviewer #4: Yes

5. Is the manuscript presented in an intelligible fashion and written in standard English?

Reviewer #3: Yes

Reviewer #4: Yes

6. Review Comments to the Author

Reviewer #3: 1.In the abstract, start off with a brief introduction to the background of the topic and pain points first. This will then give a good low towards how the research question came about and what the research aims to achieve. The objectives has to be made clearer.

2.I would suggest to highlight the increased alcohol consumption among women in the introduction as well because exploring the shifts in gender role is one of the main objectives that your study is set out to achieve.

3. What is the conceptual framework used to guide this study?

4. How was the topic guide developed?

5. Mention the type of reimbursement offered

6. Mention where did the study attain its ethical approval from.

7. Mention how was consent taken from those aged 18 years and below and was there any intervention suggested or given to minors who admitted to consuming alcohol.

Reviewer #4: Abstract:

Given the multifaceted nature of alcohol consumption in the Sabah community, I suggest the authors incorporate Bronfenbrenner's social-ecological model as a theoretical lens. This model can comprehensively capture the influences on behavior from individual to societal levels. By adopting this approach, the research can systematically analyze how factors like personal values, gender roles, family dynamics, societal perceptions, and community regulations intersect to shape alcohol consumption patterns. This comprehensive understanding will inform the development of community-centric solutions addressing multiple levels of influence, thus enhancing the effectiveness of interventions.

Introduction:

The introduction provides a thorough overview of the cultural significance and prevalence of alcohol consumption in Sabah, particularly within the Kadazandusun community. However, one major weakness is the lack of clear delineation between the historical context of alcohol use in ceremonial practices and the contemporary issues of alcohol misuse. While the introduction acknowledges the problematic implications of alcohol consumption, it does not sufficiently connect these issues to the broader socio-cultural dynamics of Sabah. Additionally, the introduction could benefit from a clearer articulation of the research gap or question that the study aims to address. Establishing this gap would provide a clearer context for the significance of the research and guide the reader towards the study's objectives.

Literature Review:

The literature review presents a broad overview of existing studies on alcohol consumption patterns in Sabah, Malaysia. However, it lacks critical engagement and synthesis of the literature, which diminishes its effectiveness. While summarizing the findings of various studies, the review fails to analyze the strengths and limitations of each study or to identify potential gaps or contradictions in the literature. A more critical approach would involve evaluating the methodological rigor of the studies, discussing the implications of their findings, and synthesizing the literature to highlight areas where further research is needed. Additionally, the review would benefit from a clearer articulation of how each study contributes to understanding alcohol consumption in Sabah specifically. I suggest the authors revise the literature review to provide a more in-depth analysis of existing research, critically evaluate the strengths and weaknesses of each study, and identify gaps in the literature that their study aims to address. This will enhance the overall quality and relevance of the literature review to the current study's objectives.

Participants:

The major weakness of this section lies in the lack of detail regarding the selection criteria for participants and the rationale behind choosing the specific study areas. The section would benefit from providing more information on why these four zones were selected and how they represent the broader population of the western coast in terms of cultural dynamics and alcohol abuse. Additionally, while the age range of participants is mentioned, there is no explanation of how participants were recruited within this age range or whether any specific criteria were used to ensure diverse perspectives. To improve this section, the authors could provide a clearer rationale for the selection of study areas and participants, outline any specific inclusion criteria, and detail the recruitment process to ensure transparency and rigor in participant selection. Furthermore, including information on any efforts made to ensure diversity within the focus groups, such as considering gender, socioeconomic status, or ethnicity, would strengthen the overall methodology.

Procedure:

The major weakness of this section lies in its brevity and lack of detail regarding the procedure followed during the focus group discussions. While it outlines the general setup and duration of the interviews, it lacks specific information on the facilitation process, the structure of the discussions, and any measures taken to ensure the validity and reliability of the data collected. Additionally, there is no mention of any efforts made to address potential biases or power dynamics within the focus groups, such as ensuring equal participation among all participants or mitigating social desirability bias. To improve this section, the authors could provide a more comprehensive description of the facilitation process, including how the facilitator encouraged open dialogue and ensured all voices were heard. They should also detail any strategies used to maintain confidentiality and obtain informed consent from participants. Furthermore, discussing how the interview questions were developed and piloted could enhance the transparency and credibility of the research methodology. Finally, including information on how the data were analyzed, such as through thematic analysis or qualitative coding, would provide insight into the rigor of the research process.

Reflexivity:

This section on reflexivity effectively acknowledges potential biases and challenges inherent in the research process, such as the urban-centric perspective and the researcher's background. It demonstrates a thoughtful approach to addressing these issues through the use of reflective diaries and regular discussions with supervisors. These measures aim to enhance the credibility of the study by minimizing the impact of biases on data interpretation. Overall, the section appears well-considered and does not necessarily require revision. However, if the authors wish to provide additional detail on how reflexivity was integrated throughout the research process or how specific biases were identified and addressed, they could consider expanding this section for greater clarity and transparency.

Ethical consideration:

This section effectively addresses the ethical considerations related to participant confidentiality and the potential for community backlash. It outlines the measures taken to protect participants' identities, such as using alphabetical codes and omitting specific details about locations. By implementing these strict measures, the researchers aimed to build trust and create a safe environment for participants to share their experiences openly. Overall, the section appears well-written and does not necessarily require revision. However, if the authors wish to provide additional detail on how participant confidentiality was maintained throughout the research process or how participants were informed about these measures, they could consider expanding this section for further clarity.

Data analysis:

Strengths:

• Rigorous Transcription Process: The use of a third-party assistant with specialized training for accurate verbatim transcription minimizes potential biases and ensures the reliability of the data.

• Comprehensive Analytical Approach: The combination of manual thematic analysis and the utilization of ChatGPT for data organization demonstrates a thorough and systematic approach to data analysis.

• Iterative Code Development: The collaborative effort within the research team to develop and refine codes ensures that the analysis is grounded in the data and reflects the complexity of participants' experiences.

• Expert Review Process: Seeking feedback from seasoned researchers and community representatives enhances the credibility of the analysis and strengthens the validity of the findings.

Weaknesses:

• Lack of Detail on Analytical Techniques: While the section outlines the general analytical process, it could benefit from providing more explicit details on the specific techniques used during thematic analysis and how ChatGPT was employed to support the manual analysis.

• Limited Explanation of Peer Debriefing Sessions: Although mentioned, the section does not elaborate on how peer debriefing sessions were conducted or how they contributed to the analytical process.

Suggestions for Improvement:

• Provide More Detail on Analytical Techniques: The authors can enhance this section by explicitly outlining the steps involved in thematic analysis, including how themes were identified, coded, and refined. Additionally, they should explain in more detail how ChatGPT was utilized to support the manual analysis process.

• Elaborate on Peer Debriefing Sessions: The authors should provide a more detailed description of how peer debriefing sessions were conducted, including who participated, the frequency of sessions, and how feedback was incorporated into the analysis. This will strengthen the transparency and credibility of the analytical process.

Result

The Cultural Backbone and Its Evolution:

While the analysis under this theme adequately acknowledges the historical significance of alcohol and its evolving role within the community, it falls short in providing a deeper exploration of the underlying drivers behind the observed shifts in consumption patterns. The narrative touches upon the impact of external factors like the COVID-19 pandemic but lacks a comprehensive examination of the socio-economic, cultural, and psychological forces influencing these changes. Furthermore, there is a notable oversight in the analysis of underage drinking, with insufficient exploration of the factors contributing to its normalization and the potential repercussions for individuals and the community at large. Addressing these shortcomings would enhance the depth and criticality of the analysis, offering a more nuanced understanding of the complex dynamics surrounding alcohol consumption within the Sabah community.

The Double-Edged Sword of Visibility and Accessibility:

The analysis of findings under this theme adequately portrays the shift in alcohol availability and accessibility within the community, highlighting the ease with which alcohol can now be obtained compared to previous years. However, there is a lack of critical examination regarding the implications of this increased accessibility on alcohol-related behaviors and societal norms. The analysis fails to delve into the potential consequences of widespread alcohol availability, such as increased rates of alcohol misuse, underage drinking, or alcohol-related harm. Furthermore, while the narrative touches upon the affordability of alcohol, there is limited exploration of how changes in pricing may impact consumption patterns and socioeconomic disparities within the community. By addressing these gaps, the analysis could offer a more comprehensive understanding of the complex dynamics surrounding alcohol accessibility and its implications for community well-being.

Navigating Social Influences and Evolving Gender Norms in Alcohol Consumption:

The analysis of findings under this theme effectively highlights the influence of social constructs, such as peer pressure and evolving gender norms, on alcohol consumption patterns within the community. However, there is a notable absence of deeper exploration into the underlying reasons for these shifts in social dynamics and their broader implications. For instance, while the narrative touches upon the increasing participation of women in alcohol consumption and production, there is limited discussion on the societal factors driving this change or the potential consequences for gender equality and social norms. Additionally, while family influence is acknowledged, the analysis could benefit from a more nuanced examination of how intergenerational transmission of drinking habits intersects with broader social and cultural factors to shape attitudes and behaviors toward alcohol. By addressing these gaps, the analysis could provide a more comprehensive understanding of the complex interplay between social influences and alcohol consumption within the community.

The Multifaceted Impact of Alcohol on Life's Dimensions:

The analysis of findings under this theme effectively captures the multifaceted impact of alcohol consumption on various dimensions of life, including health, relationships, work, and financial stability. However, there is a potential weakness in the analysis concerning the depth of exploration into the underlying factors contributing to these impacts and the complexities of their interplay. While the narratives provide valuable insights into individuals' experiences and perceptions, there is a lack of discussion on broader societal factors that may exacerbate or mitigate these effects. For example, the analysis could delve deeper into the structural determinants of alcohol-related harm, such as socioeconomic disparities, access to healthcare, and cultural norms surrounding alcohol use. Additionally, while individual anecdotes shed light on the personal consequences of alcohol consumption, there is limited examination of systemic issues, such as alcohol marketing practices, regulatory policies, and healthcare interventions, which may influence patterns of consumption and associated outcomes. By incorporating a more comprehensive analysis of these contextual factors, the discussion could offer a richer understanding of the complex dynamics underlying the impact of alcohol on individuals and communities.

Discussion:

While the discussion effectively synthesizes the findings of the study with existing literature, it falls short in explicitly demonstrating the uniqueness of the findings compared to previous studies. Although it references broader trends and insights from prior research, it lacks a clear delineation of how the current study's findings diverge or contribute novel insights to the existing body of knowledge. To strengthen the discussion, it would be beneficial to explicitly highlight any novel findings or unique aspects uncovered by the study that distinguish it from previous research. This could involve identifying specific nuances or patterns in alcohol consumption behaviors within the Sabah community that have not been extensively documented in prior studies. By emphasizing the distinctiveness of the findings, the discussion could further underscore the significance of the study and its contribution to advancing understanding in the field of alcohol consumption research.

Conclusion:

The conclusion effectively summarizes the key findings and emphasizes the need for a holistic approach to addressing the complex dynamics of alcohol consumption in the Sabah community. However, one potential weakness is the lack of specific recommendations or actionable steps for addressing the identified challenges. While the conclusion highlights the importance of community-based solutions and awareness-raising efforts, it would be strengthened by providing more concrete suggestions for interventions or policies that could be implemented to address the evolving patterns of alcohol consumption. Additionally, the conclusion could benefit from a brief reflection on the implications of the study's findings for future research or practice in the field of alcohol misuse prevention and intervention. By incorporating these elements, the conclusion would offer a more comprehensive and actionable framework for addressing the complexities of alcohol consumption in the Sabah community.

7. PLOS authors have the option to publish the peer review history of their article (what does this mean?). If published, this will include your full peer review and any attached files.

Reviewer #3: **Yes: **Kishwen Kanna Yoga Ratnam

Reviewer #4: No

---

## [Author Response · Author response to Decision Letter 1]

25 May 2024

The response to reviewers have been compiled and uploaded with the manuscript

---

## [Decision Letter · Decision Letter 2]

16 Jul 2024

PONE-D-23-38278R2Gendered narratives and cultural shifts: a qualitative study on decadal changes in community alcohol consumptionPLOS ONE

Dear Dr. Bhutasang,

Thank you for submitting your manuscript to PLOS ONE. After careful consideration, we feel that it has merit but does not fully meet PLOS ONE’s publication criteria as it currently stands. Therefore, we invite you to submit a revised version of the manuscript that addresses the points raised during the review process.

We look forward to receiving your revised manuscript.

Kind regards,

Gabriel Caluzzi

Guest Editor

PLOS ONE

Journal Requirements:

Additional Editor Comments:

Reviewer 1 has offered useful comments to improve this manuscript. These include increasing clarity; cohesiveness between literature, findings and conclusions; and clearly highlighted implications. I hope the authors can incorporate these without much restructuring of the paper.

Reviewers' comments:

Reviewer's Responses to Questions

**Comments to the Author**

1. If the authors have adequately addressed your comments raised in a previous round of review and you feel that this manuscript is now acceptable for publication, you may indicate that here to bypass the “Comments to the Author” section, enter your conflict of interest statement in the “Confidential to Editor” section, and submit your "Accept" recommendation.

Reviewer #4: All comments have been addressed

2. Is the manuscript technically sound, and do the data support the conclusions?

Reviewer #4: Partly

3. Has the statistical analysis been performed appropriately and rigorously? 

Reviewer #4: N/A

4. Have the authors made all data underlying the findings in their manuscript fully available?

Reviewer #4: Yes

5. Is the manuscript presented in an intelligible fashion and written in standard English?

Reviewer #4: (No Response)

6. Review Comments to the Author

Reviewer #4: Abstract Revision Suggestions:

Clearly articulate specific gendered narratives and cultural shifts in alcohol consumption.

Provide examples from qualitative data to illustrate these narratives.

Detail cultural transformations influencing drinking habits.

Explain how these shifts interact with gender dynamics.

Describe the analytical framework used for interpreting data.

Discuss economic consequences linked to gender and cultural contexts.

Highlight implications for gender-sensitive and culturally appropriate public health interventions.

Mention the theoretical framework used to engage readers.

Introduction Revision Guidance:

Define key terms and theories related to gender differences, cultural dynamics, and alcohol misuse.

Emphasize decadal changes in alcohol consumption patterns.

Clearly state specific research objectives.

Literature Review Enhancements:

Synthesize findings across studies to develop a cohesive narrative.

Explore unique cultural beliefs influencing alcohol consumption.

Critique methodological approaches and biases.

Explicitly connect research objectives to identified gaps.

Focus on gender dynamics in alcohol use within Sabah.

Methodology Section Improvements:

Diversify sampling methods for better representation.

Justify age range selection for relevance.

Detail management of dialectal variations in data collection.

Expand on strategies for minimizing researcher bias.

Discuss ethical considerations and confidentiality measures.

Provide deeper insights into ChatGPT’s role in data analysis.

Results Section Refinements:

Enhance transitions between themes for clarity.

Provide comprehensive analysis across all themes.

Integrate participant quotes evenly to support findings.

Discussion Section Strengthening:

Link findings explicitly to relevant literature and theories.

Analyze the impact of COVID-19 on alcohol consumption.

Contextualize findings within broader Malaysian or global contexts.

Offer specific recommendations for policy interventions.

Conclusion Section Enhancement:

Summarize key findings and their implications succinctly.

Clarify connections between findings and recommendations.

Reinforce relevance to academic discourse and practical applications.

By addressing these areas, each section can be strengthened to improve coherence, clarity, and scholarly rigor in discussing alcohol consumption in Sabah, Malaysia.

7. PLOS authors have the option to publish the peer review history of their article (what does this mean?). If published, this will include your full peer review and any attached files.

Reviewer #4: No

---

## [Author Response · Author response to Decision Letter 2]

8 Aug 2024

I have attached the response to reviewer word document.

---

## [Editor Report · Decision Letter 3]

12 Aug 2024

Gendered narratives and cultural shifts: a qualitative study on decadal changes in community alcohol consumption

PONE-D-23-38278R3

Dear Dr. Wider,

We’re pleased to inform you that your manuscript has been judged scientifically suitable for publication and will be formally accepted for publication once it meets all outstanding technical requirements.

Kind regards,

Gabriel Caluzzi

Guest Editor

PLOS ONE
---

## [Editor Report · Acceptance letter]

3 Sep 2024

PONE-D-23-38278R3 

PLOS ONE

Dear Dr. Wider, 

I'm pleased to inform you that your manuscript has been deemed suitable for publication in PLOS ONE. Congratulations! Your manuscript is now being handed over to our production team.

Kind regards, 

on behalf of

Dr. Gabriel Caluzzi 

Guest Editor

PLOS ONE